# Characterization of Stealth Liposome-Based Nanoparticles Encapsulating the ACAT1/SOAT1 Inhibitor F26: Efficacy and Toxicity Studies In Vitro and in Wild-Type Mice

**DOI:** 10.3390/ijms25179151

**Published:** 2024-08-23

**Authors:** Junghoon Lee, Adrianna L. De La Torre, Felix L. Rawlinson, Dylan B. Ness, Lionel D. Lewis, William F. Hickey, Catherine C. Y. Chang, Ta Yuan Chang

**Affiliations:** 1Department of Biochemistry and Cell Biology, Geisel School of Medicine at Dartmouth, Hanover, NH 03755, USA; junghoon.lee.gr@dartmouth.edu (J.L.);; 2Clinical Pharmacology Shared Resource, Norris Cotton Cancer Center, Dartmouth-Hitchcock Medical Center, Lebanon, NH 03766, USA; 3Department of Pathology, Dartmouth-Hitchcock Medical Center, Lebanon, NH 03766, USA

**Keywords:** ACAT1/SOAT1, nanoparticle, F1251, F26, cholesterol, cholesteryl ester, DSPE-PEG, phosphatidylcholine, neurodegenerative diseases, Alzheimer’s disease

## Abstract

Cholesterol homeostasis is pivotal for cellular function. Acyl-coenzyme A:cholesterol acyltransferase 1 (ACAT1), also abbreviated as SOAT1, is an enzyme responsible for catalyzing the storage of excess cholesterol to cholesteryl esters. ACAT1 is an emerging target to treat diverse diseases including atherosclerosis, cancer, and neurodegenerative diseases. F12511 is a high-affinity ACAT1 inhibitor. Previously, we developed a stealth liposome-based nanoparticle to encapsulate F12511 to enhance its delivery to the brain and showed its efficacy in treating a mouse model for Alzheimer’s disease (AD). In this study, we introduce F26, a close derivative of F12511 metabolite in rats. F26 was encapsulated in the same DSPE-PEG_2000_/phosphatidylcholine (PC) liposome-based nanoparticle system. We employed various in vitro and in vivo methodologies to assess F26’s efficacy and toxicity compared to F12511. The results demonstrate that F26 is more effective and durable than F12511 in inhibiting ACAT1, in both mouse embryonic fibroblasts (MEFs), and in multiple mouse tissues including the brain tissues, without exhibiting any overt systemic or neurotoxic effects. This study demonstrates the superior pharmacokinetic and safety profile of F26 in wild-type mice, and suggests its therapeutic potential against various neurodegenerative diseases including AD.

## 1. Introduction

Cholesterol, an essential sterol lipid, is vital for maintaining cell membrane integrity [1]. The balance of cholesterol within cells plays a pivotal role in cellular function and metabolism, and its disruption can lead to cell toxicity, and the development of multiple diseases such as atherosclerotic cardiovascular diseases, cancer, and neurodegenerative diseases [2,3].

Acyl-coenzyme A (CoA):cholesterol acyltransferases (ACATs), also known as sterol O-acyltransferases (SOATs), are key enzymes in regulating cholesterol homeostasis by converting excess cholesterol into cholesteryl esters for storage [4,5]. ACAT exists in two isotypes, ACAT1 and ACAT2, which are found in various mammalian species [6,7,8,9]. ACAT1 and ACAT2 are both membrane-spanning proteins found in the endoplasmic reticulum (ER) that play a fundamental role in cholesterol metabolism across different tissues [10]. While ACAT1 is ubiquitously expressed across nearly all human tissues, ACAT2 is predominantly found in intestinal enterocytes; the ACAT2 levels are elevated in specific diseases and conditions such as hepatocarcinoma [11,12,13]. Interestingly, in a normal, non-diseased state, the expression patterns of these enzymes in the liver differ between species; in human liver cells, ACAT1 is the dominant isoenzyme, whereas in mouse liver cells, ACAT2 is more prominently expressed [8,14,15,16,17,18]. In other mouse tissues, the ACAT1 expression dominates that over ACAT2.

Given its crucial role in cholesterol metabolism, ACAT1 has emerged as a therapeutic target for treating a broad spectrum of diseases, including atherosclerosis, cancer, dyslipidemia, and neurodegenerative diseases like Alzheimer’s disease (AD) and Niemann-Pick Type C disease (NPCD) [19,20,21,22,23,24,25,26,27,28,29,30,31]. The importance of ACAT inhibition in treatment strategies is underscored by decades of research, though only a few small molecule ACAT inhibitors such as CI-1011 (avasimibe), CS-505 (pactimibe), K-604, and F12511 (eflucimibe) have reached clinical trials for hypercholesterolemia and atherosclerosis [32,33,34,35,36]. While these clinical outcomes have only demonstrated limited efficacy, they provide the potential to target ACAT for various pathologies [37]. Recent studies have further suggested repurposing ACAT inhibitors, traditionally developed for cardiovascular diseases, for treating neurodegenerative diseases due to their role in altering lipid metabolism [19,26,28,29,38,39,40,41,42,43,44].

In earlier studies, we assessed the efficacy of F12511, a potent ACAT1 inhibitor, for its application in AD [28,45]. F12511 exhibits strong inhibition of ACATs with IC_50_ values of 39 nM for human ACAT1 and 110 nM for human ACAT2 [12,35,46]. However, the application of F12511 in vivo is constrained by its high hydrophobicity, which impedes its solubility in aqueous phase [47,48,49]. To circumvent this limitation, we utilized a liposome-based nanoparticle delivery system to encapsulate F12511. These nanoparticles, varying in size from 10 to 1000 nm, are designed to improve the solubility of hydrophobic agents, thereby increasing bioavailability and biodistribution [50]. Poly-(ethylene glycol) (PEG) serves as a critical component of these liposome-based nanoparticles by providing stability in the bloodstream and increasing their retention and uptake in target tissues, a feature referred to as “stealth liposome” [51,52,53,54,55]. Additionally, we applied this formulation and demonstrated its efficacy in alleviating amyloidopathy and tauopathy in the 3xTg AD mice [28]. 

Here, we advanced our exploration of ACAT inhibitors by characterizing F26, which is a synthetic analog derived from one of three metabolites of F12511, identified in rat plasma. F26 was identified as effective against rat liver microsomal ACAT [56,57]. Rat liver mainly expresses ACAT2; the efficacy of F26 in inhibiting ACAT1 was not studied. We used DSPE-PEG_2000_ and phosphatidylcholine (PC) to encapsulate F26 within DSPE-PEG_2000_/PC-based nanoparticles and designated them as nanoparticles encapsulating F26 (NP F26). Following encapsulation, we tested the delivery of F26 across systemic tissues and the brain of wild-type (WT) mice. Our experiments reveal that F26 not only surpasses F12511 in ACAT1 inhibitory efficacy but also exhibits no significant toxicity in both the brain and systemic tissues of WT mice. Overall, these studies highlight F26’s potential as a superior therapeutic agent to F12511 for various diseases, especially neurodegenerative diseases like AD.

## 2. Results

### 2.1. Characterization of the Potent ACAT1 Inhibitor F26

Extending our previous work with F12511, we introduce F26, a synthetic analog of F12511 metabolites found in rat serum (Figure 1) [28,45,56]. The chemical structures of F12511 and F26 are illustrated in Figure 1A. F12511 has a molecular weight of 469.72 g/mol, while F26 is slightly larger at 519.72 g/mol due to the addition of a sulfonyl group (–SO_2_) attached to the long aliphatic side chain and a fluorine atom introduced at the asymmetric carbon of F12511. Research from Pierre Fabre demonstrated that F26 (referred to as compound 3 in [57]) is more potent than F12511 for inhibiting rat liver microsomal ACAT [57]. Rat liver is similar to mouse liver and predominantly expresses ACAT2. In order to assess the efficacy of F26 in inhibiting ACAT1, we first assessed the efficacy of F26 in inhibiting cholesterol esterification by ACAT using mouse embryonic fibroblasts (MEFs); MEFs predominantly express ACAT1, not ACAT2 [26]. To evaluate and compare the inhibitory activity towards ACAT1 of F26 relative to F12511, we measured ACAT activity by [^3^H] oleate pulse assay in MEFs treated with F12511 and F26. The results showed that both F12511 and F26 inhibited ACAT activity in MEFs, displaying IC_50_ values of 20.6 nM and 3.0 nM, respectively (Figure 1B). Of particular note, the IC_50_ value for F12511 was first identified in our previous study using human ACAT1 expressed in CHO cells, with an IC_50_ of 39 nM; however, the experiment described in Figure 1B was conducted in MEFs which measures ACAT1 in mouse, which may account for the difference in IC_50_ values [12]. The data in Figure 1B indicate that F26 is more potent as an ACAT1 inhibitor compared to F12511 by close to seven-fold. Earlier studies established that liposome-based nanoparticle formulations could successfully deliver F12511 into the brain in vivo [28,45]. We explored whether F26 can also be encapsulated in DSPE-PEG_2000_/PC-based nanoparticles (vehicle) effectively. We prepared nanoparticles encapsulating F26 (NP F26) with 6 mol% (30 mM) of DSPE-PEG_2000_, 20 mol% (6 mM) of PC, and 40 mol% (12 mM) of F26 [45]. After formulating NP F26, we assessed the encapsulation efficiency using thin-layer chromatography (TLC) followed by iodine staining (Figure 1C, left). A series of F26 standards were utilized to generate a standard curve for quantifying the amount of F26 in both the supernatant and precipitated pellet of NP F26 samples (Figure 1C, right). The results showed that approximately 9 mM of F26, which is 75% of the initial amount, was encapsulated in the NP F26 supernatant, with the remainder present in the pellet. This encapsulation efficiency is comparable to that observed with nanoparticles encapsulating F12511 (NP F12511), indicating that both F12511 and F26 exhibit similar encapsulation rates when incorporated into DSPE-PEG_2000_/PC-based nanoparticles. Next, we characterized the particle size and polydispersity index (PDI) of nanoparticles using dynamic light scattering measurements via a Zetasizer Nano ZS at 1 month and 6 months post-formulation (Figure 1D). The result showed that at 1 month, the average NP F26 diameter was 224.4 nm with a PDI of 0.236, similar to that of NP F12511 [45]. At 6 months, NP F26 displayed a more dispersed distribution (PDI = 0.335) and a reduced size (187.6 nm), though the majority (93.1%, peak 1) remained the same as the initial formulation. This indicates that NP F26 can maintain its stability over 6 months. Importantly, Appendix A represented correlograms that illustrate the decay of scattering intensity, providing additional insights into the quality of NP F26. Despite NP F26 at 6 months not meeting the quality criteria, the persistence of the primary particle size peak (93.1%, peak 1) suggests significant stability over this period. To ensure adherence to quality standards, the usage of NP F26 is recommended for periods less than 6 months. The “vehicle” nanoparticles, which do not contain inhibitors, exhibited a broad particle size distribution with two major peaks in both 1-month and 6-month samples, likely representing small micelles (~15 nm) and unilamellar liposomes (~200 nm) formed by DSPE-PEG_2000_ and PC. The zeta potentials of nanoparticles, both vehicle and NP F26, were in the neutral range, owing to the PBS used for reconstitution after lyophilization.

### 2.2. NP F26 Inhibits ACAT1 More Effectively and More Durably Than NP F12511 in Cell Culture 

Previous studies have examined the potential toxicity of F12511 in both murine models and humans [35]. Additionally, we have shown that F12511 displayed no detectable toxicity at concentrations up to 10 µM in primary mouse neuronal cells [45]. Building on these findings, we conducted assays to assess the toxicity and efficacy of NP F26 compared to NP F12511 in MEFs, which have been used to assess drug cytotoxicity in cell culture (Figure 2) [58]. We first investigated the cytotoxicity by using the XTT assay with MEFs (Figure 2A). MEFs were treated with F12511, F26, vehicle, NP F12511, and NP F26 at varying concentrations (0, 50 µM, 75 µM, 100 µM, and 150 µM) for 24 h (h) and followed by measuring cell viability. The results indicated at concentrations below 50 µM, neither F12511 nor F26 affected cell viability; at 150 µM, both F26 and NP F26 began to cause measurable reductions in cell viability (81% viability). Considering the IC_50_ values of F12511 (20.6 nM) and F26 (3.0 nM) for ACAT1 inhibition, the concentrations required to induce cell toxicity (50 µM or higher) are over 1000 times higher than those needed for the effective inhibition of ACAT1. Thus, the data suggest that F26 and NP F26 can inhibit ACAT activity effectively without causing cytotoxicity, and the cytotoxicity induced by F26 at 50 µM or higher is independent of its ability to inhibit ACAT1. We also evaluated the efficacy of NP F12511 and NP F26 through a [^3^H] oleate pulse assay in MEFs, conducted in both a dose- and time-dependent manner (Figure 2B,C). The result showed that at a concentration of 400 nM, both NP F12511 and NP F26 significantly inhibited ACAT activity (by more than 75%). At 40 nM, NP F26 demonstrated greater inhibition efficacy than NP F12511, reinforcing the earlier findings demonstrating the lower IC_50_ value of F26 (Figure 2B). Furthermore, at a concentration of 40 nM, NP F26 sustained its inhibition on ACAT for over 48 h, whereas ACAT activity began to increase at 48 h with NP F12511 (Figure 2C). This suggests that NP F26 not only inhibits ACAT1 more strongly but also maintains this inhibition longer than NP F12511 after initial treatment. To investigate if F12511 and F26 can retain their inhibitory effects after their removal from the culture medium, we washed out the medium after a 2 h treatment (with vehicle, or 40 nM NP F12511, and or 40 nM NP F26, respectively), incubated in fresh medium without the inhibitor for designated times (0, 2 h, 4 h, and 8 h), and then administered a [^3^H] oleate pulse for 2 h (Figure 2D). The results showed that after a washout, the group treated with NP F12511 exhibited robust ACAT activity and engaged in active cholesteryl synthesis. We speculate that this spike in ACAT activity might be influenced by the accumulation of free cholesterol near or at the ACAT1 site; following the washout of F12511, the free cholesterol accumulated around ACAT was available to ACAT and was rapidly utilized as a substrate [59]. Our earlier study showed that F12511’s inhibition of ACAT might not be removable by washout, but our data suggest that F12511, when encapsulated in nanoparticles, may be more easily washed out than F12511 alone [59]. Additionally, our results (Figure 2D) show that in contrast to NP F12511, the NP F26-treated group retained its inhibitory effect even 8 h after washout, suggesting that F26 may have a stronger binding affinity to ACAT1 than F12511. Collectively, these findings underscore the low cytotoxicity of F26 and NP F26, and their superior efficacy in inhibiting ACAT1 over F12511.

### 2.3. NP F26 Efficacy In Vivo Assessed via IV Administration

A previous study demonstrated that DSPE-PEG_2000_/PC-based nanoparticles serve as an for ACAT inhibitors, particularly F12511, facilitating their delivery to the brain in mouse models [28]. A single IV administration of NP F12511 in wild-type (WT) mice significantly inhibited ACAT activity in various tissues, including the forebrain, cerebellum/brain stem, adrenal gland, and liver [28]. We expanded the study to include four groups: PBS, vehicle, NP F12511, and NP F26, each administered via a single intravenous (IV) injection to assess comparative efficacy and pharmacodynamics studies. We monitored ACAT activity in these tissues at 6 h, 24 h, 48 h, and 72 h post-administration (Figure 3). Figure 3A shows that a high dose (46 mg/kg) of NP F12511, and both low (17 mg/kg) and high (51 mg/kg) doses of NP F26, significantly reduced ACAT activity at 4 h post-administration in the forebrain, cerebellum/brain stem, adrenal gland, and liver. Notably, the low dose of NP F26 exhibited inhibitory effects comparable to the high dose of NP F12511, demonstrating NP F26’s potency at 4 h post-administration. This observation aligns with our prior in vitro findings, confirming that F26 is a more potent ACAT1 inhibitor than F12511 (Figure 2). Furthermore, while NP F12511 (46 mg/kg) showed its efficacy up to only 12 h post-administration in our previous study, NP F26 (51 mg/kg) continued to display significant efficacy, reducing ACAT activity in the forebrain and cerebellum/brain stem up to 72 h after administration (Figure 3B–D) [28]. This sustained activity highlighted the enhanced duration of action of NP F26 compared to both PBS- and vehicle-treated groups, as well as NP F12511. We noted that although the ACAT activity in the forebrain and cerebellum/brain stem of NP F12511-treated mice largely returned to baseline by 24 h post-injection, the ACAT activity in the adrenal gland remained significantly reduced until 48 h, similar to the NP F26 treatment. Interestingly, we observed that the vehicle group exhibited changes in ACAT activity compared to the PBS group, indicating an unexpected but notable effect of the vehicle itself. This underscores the importance of including both PBS and vehicle controls to accurately assess the specific impacts of nanoparticle formulations. By 72 h post-administration, the ACAT activity in the adrenal glands of both NP F12511 and NP F26-treated mice was altered relative to the vehicle group, suggesting that a preferential accumulation of the vehicle and ACAT inhibitors in the adrenal gland might occur. Additionally, ACAT activity in the liver was significantly reduced by both NP F12511 (up to 24 h) and NP F26 (up to 72 h), though not as pronounced as in the brain. This suggests that the mouse liver, predominantly expressing ACAT2, might be less affected by F26 [14]. Together, these findings indicate that a single IV administration of NP F26 can significantly inhibit ACAT activity for up to 72 h in the brain and for 48 h in the adrenal gland and liver, highlighting its enhanced and prolonged efficacy compared to NP F12511.

### 2.4. NP F26 Efficacy In Vivo Assessed via IP Administration

While F12511 is a potent ACAT inhibitor with demonstrated efficacy in WT mice and AD mouse models, its administration has been limited to the IV route [28]. To explore alternative administration routes, we tested intraperitoneal (IP) injections of NP F12511 (46 mg/kg) and NP F26 (51 mg/kg) in WT mice (Figure 4). The data indicated that ACAT activity in all tissues was reduced at 6 h, 24 h, and 48 h after a single IP injection of NP F26 (Figure 4A–C). Conversely, IP delivery of NP F12511 did not demonstrate evident inhibition except in the adrenal gland. This suggests that NP F26 might offer broader therapeutic applicability. 

### 2.5. F26 Is Retained Longer Than F12511 in the Brain after a Single IV Injection of NP F26 

We next explored the pharmacokinetic profile of F26 in vivo, by quantifying F26 concentrations using liquid chromatography-tandem mass spectrometry (LC-MS/MS) as described in [56] (Figure 5). A single dose of NP F26 (51 mg/kg) was administered intravenously to 3-month-old WT mice, with F26 concentrations in plasma (ng/mL) and the brain (ng/g) measured at 4 h, 12 h, 24 h, and 48 h post-administration. The plasma and brain concentration profiles over time and a summary of the pharmacokinetic parameters are depicted in Figure 5. F26 concentrations declined in both plasma and the brain over time, indicating no preferential accumulation in the brain. However, F26 showed a slower elimination rate in the brain compared to plasma, with elimination half-lives (T_1/2_) of 10.54 h and 4.75 h, respectively. This suggested a potential for long-term inhibition of ACAT by F26 in the brain. The result showed that F26 concentration is lower in the brain than in plasma (Figure 5 and Table 1); we speculate that the high concentration of NP F26 used in this experiment probably caused F26 to penetrate the blood-brain barrier (BBB), and maintained elevated brain levels over an extended period. This suggests that NP F26 may share a similar delivery mechanism with NP F12511, facilitating the entry of ACAT inhibitors into the brain [28]. We further compared the concentrations of both F12511 and F26 in plasma and brain using brain-to-plasma ratios, with the F12511 data obtained retrieved from our previous study (Table 1) [28]. The result showed that F26 exhibited a higher brain-to-plasma ratio at 4 h than F12511, suggesting that F26 crosses the BBB more rapidly than F12511. Moreover, at the 24 h time point, the average F26 concentration remained higher than that of F12511, allowing prolonged ACAT inhibition in the brain. Notably, the F26 concentration in the brain at 48 h was higher than the F12511 concentration at 24 h, indicating that F26 retains significantly longer in the brain. Overall, these findings are consistent with measurements of ACAT activity shown in Figure 3 and Figure 4, confirming that the liposome-based nanoparticle system not only effectively delivers F26 to the brain but also establishes F26 as a more durable and potent ACAT inhibitor than F12511.

### 2.6. A 2-Week Administration of NP F26 Showed No Overt Systemic or Neurotoxicities In Vivo 

Given that NP F26 significantly reduces ACAT activity in WT mice, it was crucial to assess any potential adverse effects at the tissue level in both the brain and peripheral tissues, as previously addressed with F12511 [28]. We conducted histopathological evaluations in WT mice following 2-week IV administration of NP F26 (Figure 6). NP F26 was injected intravenously every 48 h at a dose of 51 mg/kg, with PBS and vehicle also administered as controls to account for any potential effects of the vehicle itself. No overt signs of neuronal toxicity were observed in the hippocampal and cortical regions of NP F26-treated mice compared to those treated with vehicle or PBS (Figure 6A). Dark neurons, detected around the hippocampus and dentate gyrus, were identified as common histological artifacts in brain tissues stained with H&E staining, not indicative of neurotoxicity [60]. Additionally, the cerebellum (5× and cerebellar cortex (40×) showed no histological changes among the three groups, further supporting the absence of neurotoxic effects (Figure 6A,B). No changes in adrenal gland histology were observed in both vehicle- and NP F26-treated mice, indicating that the temporal change in ACAT activity driven by either vehicle or NP F26 was not associated with systemic toxicity in the adrenal gland (Figure 3 and Figure 6B). This suggests that a potential preferential accumulation of nanoparticles and ACAT inhibitors in the adrenal gland might be released more slowly than in other tissues, thereby reducing ACAT activity for a longer duration compared to tissues like the forebrain, cerebellum/brain stem, and liver (Figure 4 and Figure 5). Lastly, no hepatotoxicity was observed among the groups, as liver tissues showed no histological abnormalities (Figure 6B). Taken together, these findings confirm that long-term treatment with NP F26 does not cause overt neurotoxicity or abnormalities in systemic tissues.

## 3. Discussion

Our current studies revealed that NP F26 significantly reduced ACAT activity in the brain for extended periods—up to 72 h with IV administration and 48 h with IP administration. This prolonged activity suggests a more effective pharmacokinetic profile of NP F26, highlighted by its ability to maintain higher brain concentrations for longer durations than F12511, as evidenced by higher brain-to-plasma ratios and slower elimination rates. Additionally, the flexibility of F26 in different administration routes presents clear advantages over F12511, particularly given F12511’s limited efficacy via IP injection, especially in the brain (Figure 4). This versatility suggests the potential for F26 delivery through various modalities, including oral, intranasal, and subcutaneous routes, which could greatly enhance its clinical applicability. Further studies are required to fully validate these alternative administration routes for F26.

The results, presented in Figure 3 and Figure 4, showed the effect of the DSPE-PEG_2000_/PC-based nanoparticle (vehicle) itself on ACAT activity in WT mice after both IV and IP administration. These findings suggest that the adrenal gland, critical for steroidogenesis, retained the vehicle and ACAT inhibitors longer due to its ability to uptake and accumulate lipophilic substances such as cholesterol and phospholipids [61,62]. This characteristic makes the adrenal gland particularly susceptible to the accumulation of hydrophobic drugs like F12511 and F26, potentially leading to toxicity [63]. To fully understand these regional effects and potential long-term accumulations of substances, further detailed studies using LC-MS/MS will be necessary. We assessed the safety of NP F26 through histopathological examinations after repeated IV injections administered every 48 h for 2 weeks. These evaluations confirmed the absence of neurotoxicity or systemic toxicity in treated mice, including the forebrain, cerebellum, adrenal gland, and liver (Figure 6). This safety profile aligns with findings from Phase 1 clinical trials for F12511, supporting the potential of NP F26 for long-term therapeutic use without adverse effects associated with prolonged exposure to certain other ACAT inhibitors [35,63].

Previous research showed the limited brain bioavailability of ACAT inhibitors like K604 and F12511 due to their poor solubility, restricting their therapeutic application toward brain-related diseases [45,64]. Advancing this work, our study employs the same liposome-based nanoparticle system to encapsulate F12511 and F26. The results indicate that upon IV delivery, F26 maintained longer and more retention in the brain than F12511 (Table 1). We also examined the efficacy and safety study of NP F26, demonstrating its robust capability in inhibiting ACAT1 activity in vivo, with a notable enhancement in duration and potency. The potential of NP F26 will next be tested in various animal and cell culture models for various medical conditions. 

The development of liposome-based nanoparticles has expanded largely since the early 2000s [65]. These formulations are highlighted for reducing drug toxicity and improving biological compatibility; they are biodegradable and non-toxic [66]. Research advancements in liposome-based nanoparticles include enhancing targeted drug delivery by conjugating antibodies, aptamers, peptides, and other small molecules into liposomes [67,68,69,70,71]. Continuous advancements in liposome-based nanoparticles could further optimize the delivery of ACAT inhibitors like F26 for treating various diseases at targeted tissues.

## 4. Materials and Methods

### 4.1. Ethical Handling of Animals

All experiments were approved by the Institutional Animal Care and Use Committee (IACUC) at Dartmouth College under protocol #00002020. Three-month-old wild-type (WT) C57BL/6 mice (from Jackson Laboratory, Bar Harbor, ME, USA and Charles River Laboratories, Wilmington, MA, USA) were randomized and assigned to various experimental groups. Animals were housed in a standard 12 h light/dark cycle with ad libitum access to food and water. The mice were subjected to treatment by an intravenous tail vein injection, retro-orbital sinus injection, and intraperitoneal injection, depending on the specific experiment. For certain experiments, the mice were intravenously administered with either NP F12511 (46 mg/kg) or NP F26 (17 mg/kg and 51 mg/kg). 

### 4.2. Materials

DSPE-PEG_2000_ was from Laysan Bio, Inc. (mPEG-DSPE, MW 2000, Arab, AL, USA). L-α-phosphatidylcholine (from egg yolk) was from Sigma-Aldrich (Catalog No. P-2772, St. Louis, MO, USA). F12511 and F26 with purity (>98%) were custom synthesized by WuXi AppTec (Shanghai, China) [35,57]. The thin layer chromatography (TLC) plates were from Analtech (Newark, DE, USA).

### 4.3. Nanoparticle Formation

The protocol is described in detail in [45,72,73,74]. In brief, DSPE-PEG_2000_ dissolved in ethanol and phosphatidylcholine (PC) dissolved in chloroform were mixed while vortexing. ACAT inhibitors (F12511 or F26) were dissolved in ethanol in the desired concentration and slowly added to the DSPE-PEG_2000_/PC mixture while vortexing. The final solution contained 30 mM DSPE-PEG_2000_, 6 mM PC, and 4–12 mM ACAT inhibitors. The final solution was then lyophilized overnight at −40 °C and stored at −20 °C until it was ready to be solubilized. The dried sample was reconstituted in 1 mL of phosphate-buffered saline (PBS) and bath sonicated in a Branson 2510 sonicator (Branson Ultrasonics, Brookfield, CT, USA) at −4 °C for 20 min intervals until no visible precipitate remained. The sample was then transferred to sterile Eppendorf tubes and centrifuged at 12,000 rpm for 5 min. The supernatant and the pellet were separated into new tubes, and the encapsulation efficiency of ACAT inhibitors was quantified. All samples were then sealed and stored at 4 °C until further use.

### 4.4. Nanoparticle Characterization

The encapsulation efficiency was determined by TLC as previously described [45]. Five (5) µL of each sample was loaded on a TLC plate. The solvent system used was hexanes:ethyl ether:acetic acid (60:40:1) [40]. The retention factor (Rf) was approximately 0.39 for F26. F26 bands were visualized by iodine staining and their intensities were measured using ImageJ software (Version 1.53f). The encapsulation efficiency of F26 was calculated by extrapolation from the standard curve of a gradient F26 standard.

The particle size and polydispersity index (PDI) of DSPE-PEG_2000_/PC nanoparticles (vehicle) and nanoparticles encapsulating F26 (NP F26) were characterized using a Zetasizer Nano ZS (Malvern Instruments, Worcestershire, UK) with Zetasizer 7.1 software. All samples were in PBS and loaded into a 10 mm path-length disposable cuvette. All samples were measured in triplicate.

### 4.5. Cell Culture

MEFs were isolated according to the procedure described in [75]. MEFs were grown as monolayers and maintained at 37 °C in a humidified condition with 5% CO_2_ in DMEM (Corning, Manassas, NY, USA) supplemented with 10% calf serum (R&D Systems, Flowery Branch, GA, USA), MEM non-essential amino acids (Gibco, Grand Island, NY, USA), and penicillin/streptomycin (Corning, Manassas, VA, USA).

### 4.6. XTT Assay

An XTT cell proliferation assay kit (measures mitochondrial redox activity in metabolically active cells) was purchased from ATCC (Catalog No. 30-1011K, Manassas, VA, USA). Cell toxicity assessment was performed according to the manufacturer’s protocol. MEFs were seeded in a 96-well plate, 2500 cells per well, and cultured in full growth medium for 24 h. The medium was then removed and replaced with 0.1 mL of fresh DMEM containing either PBS (or ethanol), vehicle, F12511 (dissolved in ethanol), F26 (dissolved in ethanol), NP F12511, and NP F26 with various concentrations for 24 h. After the treatment, XTT reagent was added to each well, and the plates were incubated at 37 °C for 4 h. Following the incubation, absorbance was measured by using a BioTek Synergy Neo2 Hybrid Multimode Reader (BioTek, Winooski, VT, USA).

### 4.7. Measuring ACAT Activity by [^3^H] Oleate Pulse Assay

The [^3^H] oleate pulse assay was performed as described [76,77]. In brief, MEFs were grown in 6-well plates at least 48 h before treatment. At 90% cell confluency, cells were treated with the following treatment groups: PBS, vehicle, NP F12511, and NP F26 for 2 h. For the washout assay, cells were washed and incubated with a conditioned drug-free medium for a given amount of time. After the incubation, cells were pulsed with 10 µL of 10 mM [^3^H] oleate/BSA (containing 4 µCi) for 2 h in a 37 °C water bath with 5% CO_2_. At the end of the pulse period, cells were placed on ice, washed three times with cold PBS, and lysed with 0.2 M NaOH. Lysates were aliquoted first for protein measurement and the remaining extract was neutralized to pH 7.0 by adding 3 M HCl and 1 M KH_2_PO_4_ at pH 7. Lipids were then extracted with chloroform:methanol (2:1) and water. Samples were vortex-mixed and centrifuged at 500 rpm for 10 min. The top phase was removed, and the bottom phase was blow-dried by nitrogen gas. Dried samples were dissolved in ethyl acetate and spotted on a thin-layer chromatography (TLC) plate (Miles Scientific Silica gel HL, Catalog No. P46911). The solvent system used was petroleum ether:ethyl ether:acetic acid (90:10:1). The cholesteryl ester band with an Rf value at 0.9 was visualized by iodine staining, scraped from the plate, and measured with a scintillation counter.

### 4.8. Measuring ACAT Activity by Mixed Micelle Assay

The mixed micelle assay was performed as described [40,46]. WT mice with a C57BL/6 genetic background were injected intravenously or intraperitoneally with either PBS, vehicle, NP F12511, or NP F26. At the indicated time post-injection, mice were euthanized by CO_2_ overload. Mouse tissues (forebrain, cerebellum/brain stem, adrenal gland, and liver) were collected and homogenized in a buffer (2.5% 3-((3-Cholamidopropyl) dimethylammonio)-1-propanesulfonate (CHAPS) and 1 M KCl in 50 mM Tris at pH 7.8) using the Next Advance Bullet Blender homogenizer with stainless-steel beads at 4 °C. Tissue homogenates were then aliquoted into chilled glass tubes that contained the mixed liposomal mixture of 9.3 mM taurocholate, 10.8 mM phosphatidylcholine (from egg yolk), and 1.6 mM cholesterol. The samples were vortex-mixed and then incubated in a 37 °C shaking water bath with 10 nmol ^3^H-oleoyl CoA/10 nmol BSA added to start the enzyme reaction for 10 min. The assay was quenched by adding chloroform:methanol (2:1) and water. Samples were vortex-mixed and centrifuged at 500 rpm for 10 min. The top phase was removed, and the bottom phase was blow-dried by nitrogen gas. Dried samples were loaded on a TLC plate with a solvent system of petroleum ether:ethyl ether:acetic acid (90:10:1). The cholesteryl ester band was collected and measured by a scintillation counter. Control experiments showed that there is no gender difference in ACAT enzyme activity in WT mice aged 3–4 months.

### 4.9. LC-MS/MS

F26 concentrations in mouse plasma and tissues were quantified via LC-MS/MS, using F12511 as the internal standard. After intravenous administration of F26, mice were perfused with PBS at scheduled time points. Plasma and brain tissue were collected, flash-frozen in liquid nitrogen, and stored at −80 °C until analysis. F26 and F12511 were dissolved in ethanol and DMSO, respectively, and stored at −40 °C. Subsequent working solutions of F26 and F12511 were made fresh daily by diluting in 80:20 methanol:water. Calibrator and quality control (QC) samples were prepared in the appropriate C57BL/6 mouse matrix: pooled plasma (anticoagulant: K3-EDTA, Innovative Research) and brain homogenate. The brain tissues were homogenized at 0.1 g/mL in deionized water using stainless-steel beads and a Next Advance Bullet Blender. Samples were cleaned up and analytes were extracted using protein precipitation followed by nitrogen dry down and reconstitution. In short, internal standard was added (plasma: 6 µL of 200 ng/mL F12511, brain: 6 µL of 100 ng/mL F12511) to samples (50 µL) followed by 175 µL of 0.1% NH_4_OH in acetonitrile. Samples were vortex-mixed for 5 min then centrifuged for 10 min at 21,130× *g* and a total of 200 µL of clear supernatant was transferred to new tubes and dried under nitrogen in a water bath at 45 °C. The dried residues were reconstituted in 80:20 methanol:water (plasma: 400 µL, brain: 200 µL), vortex-mixed for 15 s and centrifuged at 21,130× *g* for 5 min. Finally, 100 µL of supernatant was transferred to amber autosampler vials, and 2 µL was injected into the LC-MS/MS system. For the quantitative analysis, each sample was prepared in triplicate and each replicate was injected onto the LC-MS/MS system three times. Liquid chromatography was performed on a Thermo Scientific Vanquish Flex UHPLC system using buffers A = 0.1% formic acid in water and B = 0.1% formic acid in methanol. A Thermo Scientific Accucore C18 50 × 2.1 mm 2.6 µm column fitted with a 10 × 2.1 mm 2.6 µm C18 guard cartridge maintained at 40 °C was used. Separation was achieved with gradient conditions at a flow rate of 0.8 mL/min using the following steps: 80% B from 0 to 0.25 min, 80–95% B from 0.25 to 3.0 min, 95% B from 3.0 to 3.5 min, and 80% B from 3.5 to 5.0 min (re-equilibration). A Thermo Scientific TSQ Quantis triple quadrupole mass spectrometer was operated in positive ion mode with a collision pressure of 1.5 mTorr for quantification. Multiple reaction monitoring (MRM) was used to measure the following m/z transitions for quantification: 520.29→266.125 *m*/*z* (collision energy 24.12, S-Lens 139) for F26 and 470.31→268.196 *m*/*z* (collision energy 17.09, S-Lens 159) for F12511. The heated electrospray ionization (H-ESI) source was operated with a spray voltage of 3500 V, vaporizer temperature 400 °C, ion transfer tube temperature 300 °C, sheath gas 23, aux gas 12 and ion sweep gas 0.5 (gases are arbitrary units). The quantitative range for F26 was 0.5–500 ng/mL for plasma samples and 0.3–200 ng/mL for brain samples. QC samples were prepared at 1.5, 150, and 400 ng/mL for plasma and at 0.9, 80, and 150 ng/mL for brain.

### 4.10. Histology

Three-month-old WT mice, C57BL/6 background, were treated with either PBS, DSPE-PEG_2000_/PC nanoparticle (vehicle), or nanoparticles encapsulating F26 (NP F26). Each group received an intravenous (IV) injection every 48 h for 2 weeks (51 mg F26/kg and equivalent volume for PBS and vehicle). After 24 h from the last dose, animals were anesthetized with Avertin (Sigma Aldrich, St. Louis, MO, USA) and transcardially perfused with 4% paraformaldehyde (PFA) in PBS containing 4% sucrose for fixation. The tissues were collected and incubated in 4% PFA at 4 °C overnight. Fixed tissues were embedded in paraffin, and specimens were cut into serial sections of 4 µm thick. Histological characteristics were examined after hematoxylin–eosin (H&E) staining. All images were assessed using ImageJ software.

### 4.11. Data Analysis and Visualization

Data compiling, statistical analysis, and all graph visualization were created in GraphPad Prism 10 and ImageJ software. 

## 5. Patents

Dartmouth College has filed a U.S. Provisional Application entitled “Method For Attenuating Neuroinflammation, Amyloidopathy and Tauopathy”.

## Figures and Tables

**Figure 1 ijms-25-09151-f001:**
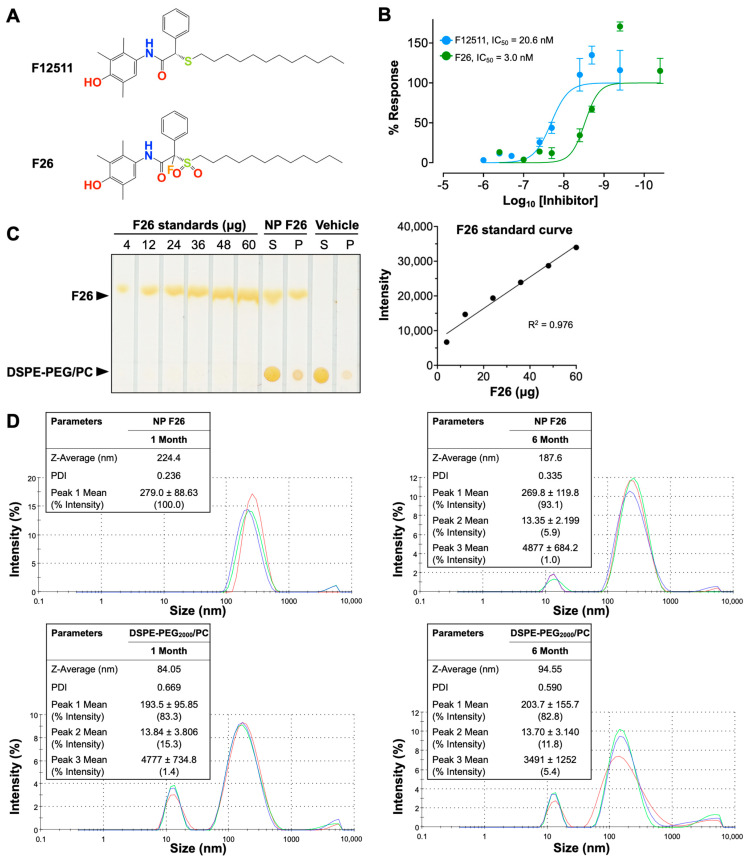
Characterization of F26 and DSPE-PEG_2000_/PC nanoparticles encapsulating F26 (NP F26). (**A**) Chemical structures of F12511 (**top**) and F26 (**bottom**). (**B**) Dose–response curves of F12511 and F26 against ACAT in MEFs. The lines show the non-linear fit used to determine the IC_50_. Each data point represents mean ± SD. (**C**) F26 encapsulation efficiency in NP F26 was analyzed by TLC with iodine staining (**left**). The standard curve obtained by plotting the F26 gradient standard (**right**) and encapsulation efficiency was calculated by using the intensity of the supernatant of NP F26. (**D**) NP F26 and DSPE-PEG_2000_/PC nanoparticles were analyzed by a Zetasizer Nano ZS at 1 and 6 months after formulation. All experiments were performed in triplicate, with each measurement represented by a different color line.; peaks are expressed as mean ± SD, and Z-Average and PDI are shown with mean values in the inserted tables. The peak order is by the area under the peak.

**Figure 2 ijms-25-09151-f002:**
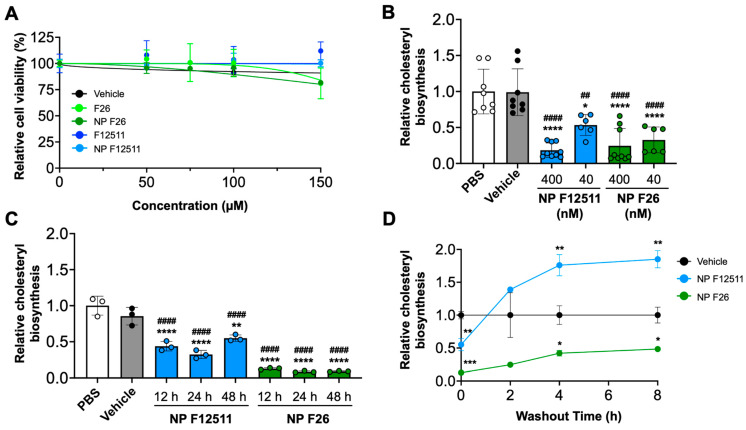
NP F26 inhibits ACAT1 more strongly and for a longer duration than NP F12511. (**A**) MEFs treated with graded concentrations of ACAT inhibitors as indicated and nanoparticles were assessed for cytotoxicity via XTT assay. Each data point represents mean ± SEM. (**B**) Dose-dependent treatments of NP F12511 and NP F26 in MEFs. Cells were treated for 2 h with indicated conditions and then pulsed for another 2 h with ^3^H-oleate to measure ACAT activities. Bars represent mean ± SD. *p*-value determined using one-way ANOVA followed by Tukey’s multiple comparisons test. (**C**) Time-dependent treatments of NP F12511 and NP F26 in MEFs. Cells were treated with 40 nM concentration for indicated time and then pulsed for another 2 h with ^3^H-oleate to measure ACAT activity. Bars represent mean ± SD. *p*-value determined using one-way ANOVA followed by Tukey’s multiple comparisons test. (**D**) MEFs were treated with 40 nM concentration for 2 h with indicated conditions and then washed with drug-free media and incubated for given amounts of time as indicated. ACAT activity was measured by ^3^H-oleate pulse for 2 h. Each data point represents mean ± SD. *p*-value determined using two-way ANOVA followed by Tukey’s multiple comparisons test. # showed statistical significance compared to PBS, and * showed statistical significance compared to vehicle. * *p* < 0.05, ##,** *p* < 0.01, *** *p* < 0.001, ####,**** *p* < 0.0001.

**Figure 3 ijms-25-09151-f003:**
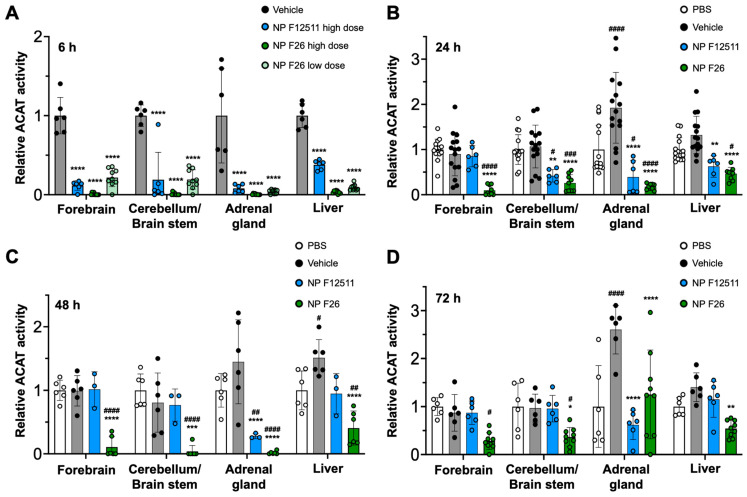
F26 inhibits ACAT activity in the forebrain, cerebellum/brain stem, adrenal gland, and liver after a single IV administration of NP F26 to WT mice at (**A**) 6 h, (**B**) 24 h, (**C**) 48 h, (**D**) 72 h time points. WT mice were injected with either PBS, vehicle, NP F12511 (46 mg/kg, high dose), or NP F26 (17 mg/kg, low dose, or 51 mg/kg, high dose). Relative ACAT activity was measured by using a mixed micelle assay. (**B**–**D**) NP F12511 (46 mg/kg) and NP F26 (51 mg/kg) were treated. Bars represent mean ± SD. Each data point represents a triplicate for each mouse. (*n* = 2–5/group). *p*-value determined using two-way ANOVA followed by Tukey’s multiple comparisons test. # showed statistical significance compared to PBS, and * showed statistical significance compared to the vehicle. #,* *p* < 0.05, ##,** *p* < 0.01, ###,*** *p* < 0.001, ####,**** *p* < 0.0001.

**Figure 4 ijms-25-09151-f004:**
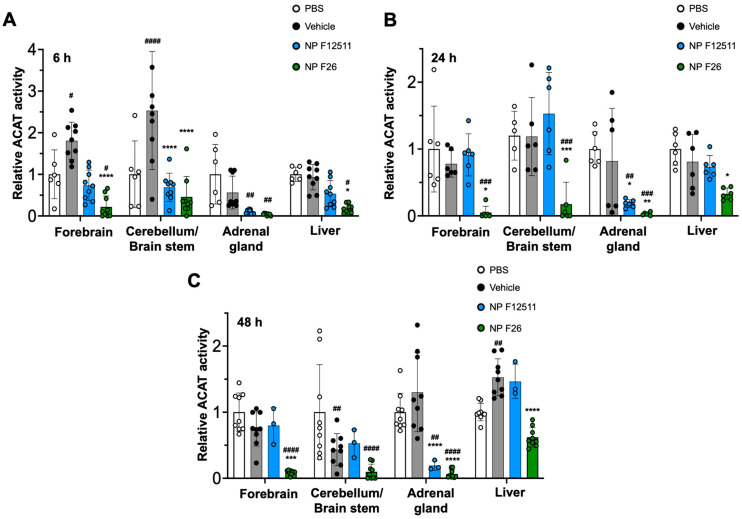
F26 inhibits ACAT activity in the forebrain, cerebellum/brain stem, adrenal gland, and liver after a single IP administration of NP F26 to WT mice at (**A**) 6 h, (**B**) 24 h, (**C**) 48 h time points, respectively. WT mice were injected with either PBS, vehicle, NP F12511 (46 mg/kg), or NP F26 (51 mg/kg). Relative ACAT activity was measured by using a mixed micelle assay. Bars represent mean ± SD. Each data point represents a triplicate for each mouse (*n* = 2–3/group). *p*-value determined using two-way ANOVA followed by Tukey’s multiple comparisons test. # showed statistical significance compared to PBS, and * showed statistical significance compared to the vehicle. #,* *p* < 0.05, ##,** *p* < 0.01, ###,*** *p* < 0.001, ####,**** *p* < 0.0001.

**Figure 5 ijms-25-09151-f005:**
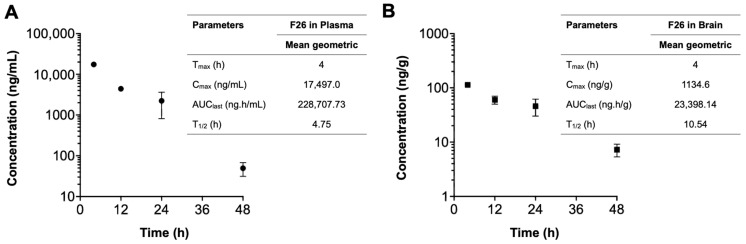
The concentration–time profiles of F26 after a single IV administration. Each plasma (**A**) and brain (**B**) sample was collected after IV administration of NP F26 at 51 mg/kg dose in three-month-old WT mice and was analyzed via LC-MS/MS. Individual points represent mean ± SD for two independent animals (*n* = 2). The pharmacokinetic analyses were performed using a non-compartmental analysis, and the parameters are shown in the inserted tables. Data are presented as mean. C_max_, maximum concentration; T_max_, time to maximum concentration; AUC_last_, area under the concentration–time curve from zero to final observation time; T_1/2_, elimination half-life.

**Figure 6 ijms-25-09151-f006:**
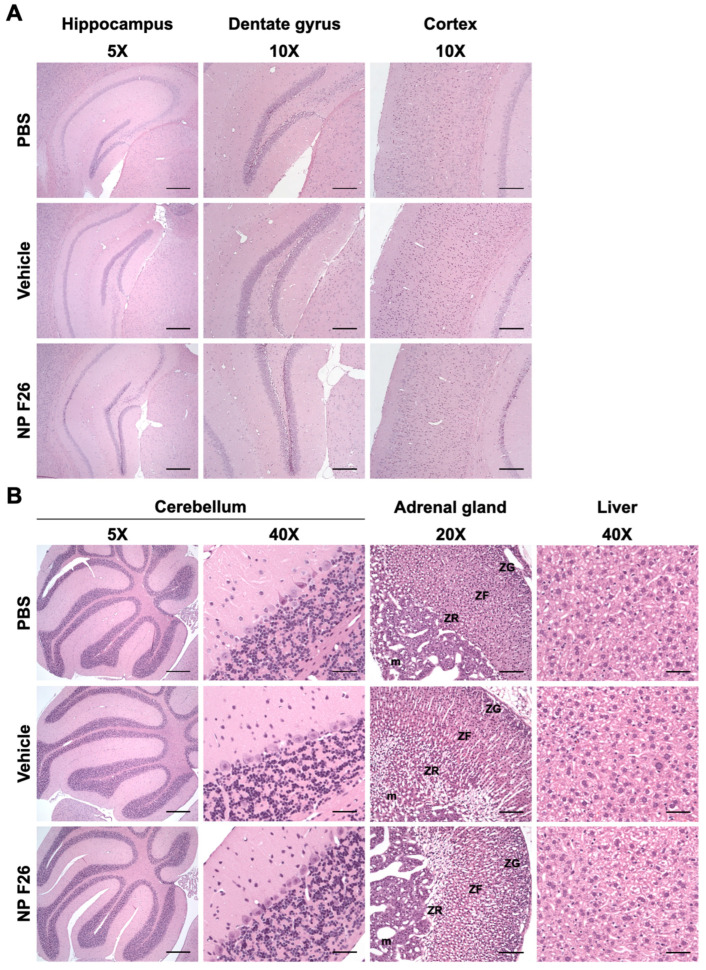
Representative images of H&E-stained histological sections of mouse tissues after 2-week IV administration of NP F26 (51 mg/kg). NP F26 treatment did not cause any histological pathology in the central nervous system and peripheral tissues compared to PBS- and vehicle-injected mouse tissues. (**A**) Images of hippocampus, dentate gyrus, and cortex. (**B**) Images of cerebellum, adrenal gland, and liver. *n* = 2/group. Abbreviations; ZG = zona glomerulosa, ZF = zona fasciculata, ZR = zona reticularis, m = medulla. Scale bars; 5× = 400 µm, 10× = 200 µm, 20× = 100 µm, 40× = 50 µm.

**Table 1 ijms-25-09151-t001:** The concentration of F26 and F12511 in plasma and brain at the indicated times after a single IV dose of NP F26 (51 mg/kg) and NP F12511 (46 mg/kg) to WT mice.

Compounds	Time	Sample Number	Plasma Concentration (ng/mL)	Brain Concentration (ng/g)	Brain-to-Plasma Ratio
F26	4 h	1	18,454	1121	0.061
		2	16,540	1148	0.069
	12 h	1	4572	671	0.147
		2	4243	529	0.125
	24 h	1	3218	571	0.178
		2	1226	347	0.283
	48 h	1	62	86	1.377
		2	37	59	1.604
F12511	4 h	1	11,185	56	0.005
		2	13,184	83	0.006
		3	12,883	98	0.008
		4	31,224	95	0.003
	12 h	1	345	25	0.072
		2	487	26	0.053
	24 h	1	101	7	0.069
		2	21	9	0.429

## Data Availability

The original contributions presented in the study are included in the article/Appendix A, further inquiries can be directed to the corresponding authors.

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
