# Peer review of "Characterization of Stealth Liposome-Based Nanoparticles Encapsulating the ACAT1/SOAT1 Inhibitor F26: Efficacy and Toxicity Studies In Vitro and in Wild-Type Mice"

_ijms, 2024, doi:10.3390/ijms25179151_

Round 1

Reviewer 1 Report

Comments and Suggestions for Authors

In the current study, the authors developed a liposome-based system to encapsulate F26, a close derivative of the previously developed metabolite F12511, targeting ACAT1 to study its efficacy and toxicities in vitro and in wild-type mice, comparing it with F12511. They found that F26 is more effective and durable than F12511 in inhibiting ACAT1 in both mouse embryonic fibroblasts (MEFs) and multiple mouse tissues, including brain tissues, without exhibiting any overt systemic or neurotoxic effects. A superior pharmacokinetic and safety profile of F26 in wild-type mice was also observed. The manuscript is well-structured and could be suitable for publication after the following revisions:

  1. From Figure 1C, it appears that the encapsulation efficiency of the liposome for F26 is almost 100%. It would be interesting to know the encapsulation efficiency of NP F26 in percentage terms.

  2. In Figure 2A, the authors compared the relative cell viability of F26 and NP F26. However, in other figures, blank F26 was not taken into account. It would be informative to compare NP F26’s efficacy with blank F26 in different assays.

  3. On page 12, lines 376-379, the authors discussed recent advancements in liposome-based nanoparticles, including enhancing targeted drug delivery using antibody-, aptamer-, peptide-, and other small molecule-conjugation into liposomes. In this context, the authors can cite a recent study demonstrating a peptide-conjugated multifunctional liposome-based platform for the early diagnosis and treatment of Alzheimer's disease (Small, 2024, p.2311670; https://doi.org/10.1002/smll.202311670).

Reviewer 2 Report

Comments and Suggestions for Authors

The author's team previously developed a stealth liposome-based nanoparticle for encapsulating the high-affinity ACAT1 inhibitor F12511, which significantly enhanced its delivery efficiency in the brain and demonstrated favourable therapeutic efficacy in a mouse model of Alzheimer's disease (AD). On this basis, this study further innovated by introducing F26, an analogue of F12511 metabolite in rats, and encapsulated F26, and systematically evaluated the differences in ACAT1 inhibitory activity, efficacy durability, and safety of F26 versus F12511 in a series of in vivo and in vivo experiments.

This study is well-designed, with scientific methodology and detailed data, which has high academic value and practical application potential.

Limited comments facilitate the authors to further improve the manuscript:

1.     Further in-depth exploration of the molecular mechanism of F26 and more extensive preclinical studies are recommended.

2.     Quality control and quality assurance of the Zetasizer Nano ZS measurement should be provided.
